# The Baluchistan Melon Fly, *Myiopardalis pardalina* Bigot: Biology, Ecology, and Management Strategies

**DOI:** 10.3390/insects16050514

**Published:** 2025-05-11

**Authors:** Junyan Liu, Yidie Xu, Mengbo Guo, Kaiyun Fu, Xinhua Ding, Sijia Yu, Xinyi Gu, Wenchao Guo, Jianyu Deng

**Affiliations:** 1Zhejiang Key Laboratory of Biology and Ecological Regulation of Crop Pathogens and Insects, College of Advanced Agricultural Sciences, Zhejiang A&F University, Hangzhou 311300, China; jliu@zafu.edu.cn (J.L.);; 2Institute of Plant Protection, Xinjiang Uygur Autonomous Region Academy of Agricultural Sciences/Key Laboratory of Integrated Pest Management on Crops in Northwestern Oasis, Ministry of Agriculture and Rural Affairs/Xinjiang Key Laboratory of Agricultural Biosafety, Urumqi 830091, China

**Keywords:** *Myiopardalis pardalina*, cucurbit, melon, invasive fly, pest controls

## Abstract

The Baluchistan melon fly is a small insect that poses a significant threat to farmers growing melons, watermelons, and cucumbers. During severe outbreaks, it can destroy up to 90% of these crops, leaving farmers with fewer products to sell and less income to live on. Our work brings together the latest research on this fly—how it lives, where it spreads, and how to control it. We demonstrate various ways to fight the fly. Farmers can change planting methods, cover fruits to keep flies away, use sprays to kill them, release natural enemies, or even apply scientific methods to make crops mor resistant. The best approach is to mix these methods into what is called integrated pest management. It works effectively and is more environmentally friendly. By collecting all this information, our review helps people understand the fly and find better ways to protect their crops. This matters because it keeps our food supply safe and supports farmers across the world.

## 1. Introduction

Invasive insect species impose profound ecological and economic burdens globally [1,2,3], with annual costs attributed to these pests estimated at USD 70 billion [4,5,6]. Among the most destructive invaders are fruit flies (Diptera: Tephritidae), a family encompassing over 5000 species [7,8,9], of which approximately 200 are of significant economic importance [9]. Over 250 tephritid species are now classified as potential quarantine threats to the European Union [7]. Similarly, Australia, China, New Zealand, Africa, South America, Asia, and North America list these insects among their top-regulated pests [10,11]. Fruit flies cause devastating crop losses, often exceeding 80% in fruit-producing regions [12]. These pests pose a critical barrier to horticultural trade and productivity, demanding urgent attention from policymakers and agricultural managers [13].

Cucurbit crops, vital to agriculture in warm climates, are particularly vulnerable to tephritid pest species. Melon (*Cucumis melo*), a key crop in the Cucurbitaceae family, is cultivated across 105 countries, covering 1.1 million hectares and yielding 28.5 million tonnes annually [14,15]. Asia dominates production, contributing 75% of cultivated land and 83% of output [16]. Alongside watermelon (*Citrullus lanatus*), cucumber (*C*. *sativus*), and pumpkin (*Cucurbita maxima*), melons support diets and economies in warm regions [17,18]. However, the Baluchistan melon fly, *Myiopardalis pardalina* (Bigot, 1891), has emerged as a highly invasive tephritid pest. It threatens global cucurbit production, which underpins both local livelihoods and international markets.

Infestations by *M*. *pardalina* result in crop losses ranging from 15 to 90%, with severe outbreaks annihilating entire harvests [19,20,21]. These losses destabilise supply chains, increase consumer prices, and threaten food security in regions dependent on cucurbits for nutrition and income. The challenges are compounded by *M*. *pardalina*’s capacity to overwinter in sub-zero temperatures [22] and its expanding range—driven by trade or natural dispersal into North America and Southern Europe [23]. Conventional insecticides are often ineffective against its internal-feeding larvae, highlighting the need for innovative, integrated management strategies.

Despite its destructive potential, *M*. *pardalina* has received relatively limited attention to date. This review synthesises current knowledge on the species’ biology, ecology, and control methods, evaluating cultural, chemical, biological, and genetic interventions. By addressing critical research gaps, we aim to provide researchers, policymakers, and agricultural practitioners with insights to develop sustainable solutions. Protecting cucurbit crops from *M*. *pardalina* is not solely an agricultural priority but also a crucial step in safeguarding rural economies, cultural traditions, and global food security amidst escalating environmental and economic uncertainties.

## 2. Systematic Literature Review

The Baluchistan melon fly *M*. *pardalina* belongs to the kingdom Animalia, phylum Arthropoda, class Insecta, order Diptera, and family Tephritidae. The genus *Carpomya* is a synonym of *Myiopardalis* [24], which explains the dual nomenclature of “*Myiopardalis pardalina*” and “*Carpomya pardalina*” in the literature. To conduct a comprehensive review, we performed a systematic search using the search string (Carpomya OR Myiopardalis) AND (pardalina) on Scopus, targeting titles, keywords, and abstracts, which yielded 13 relevant papers. In parallel, an advanced search on Google Scholar was executed by entering “pardalina” in the “with all the words” field and “Carpomya Myiopardalis” in the “with at least one of the words” field, with results filtered to include occurrences anywhere in the articles. This search returned 206 papers. After removing duplicates and evaluating relevance, 98 unique and pertinent papers were retained for analysis.

The use of Scopus and Google Scholar was international, as these platforms complement each other in scope. Scopus primarily indexes peer-reviewed academic articles from commercial publishers, whereas Google Scholar encompasses a broader range of sources, including both the academic and grey literature. The grey literature—defined as information produced and distributed by governmental, academic, business, and industrial entities outside traditional commercial publishing channels—has gained recognition for its value in systematic reviews and meta-analyses [25,26]. By incorporating the grey literature, our review captures recent and interdisciplinary research on *M*. *pardalina* that may not yet have been indexed in commercial databases, ensuring a more thorough and relevant synthesis of knowledge.

## 3. Overview of the Baluchistan Melon Fly

### 3.1. Morphological Characteristics

The life stages of *M*. *pardalina* exhibit distinct morphological traits (Figure 1). Eggs are elliptical, glossy, white, and measure 1.2 × 2.0 mm (Figure 1a). Larvae, reaching approximately 10 mm in length, are cream-white and apodous [27] (Figure 1b). Pupation yields coarctated, brown pupae averaging 7.2 mm in length [27,28,29] (Figure 1c). Adults exhibit sexual dimorphism: males possess a body length of 5.0–6.4 mm with wings spanning 4.0–4.6 mm, whereas females are larger, measuring 6.3–7.3 mm in body length with wings extending 4.5–5.3 mm [30] (Figure 1d).

The head is dark yellow, broader than it is long, and lacks facial spots or silvery markings on the frons and parafrons. A flat or convex face features distinct antennal grooves and tubercles, accompanied by elongated compound eyes. Antennae are shorter than the facial length, paired with a short, capitate proboscis. The mesonotum ranges from light yellow to brown, adorned with five black lateral spots and a central black spot at the posterior basal margin. The scutellum is light yellow, bearing a small medial black dot on its disc. Legs are unmarked by dark femoral maculations. The abdomen, yellow to orange-brown, comprises separate tergites; tergites III–V lack dark median longitudinal stripes, and tergite V is devoid of glandular spots. Males lack dark setae on tergite III, while females exhibit an exposed tergite VI equal in length to tergite V [31].

Wings are light yellow, with banding patterns akin to *Rhagoletis*, featuring basal, median, and pre-apical crossbands that extend to the posterior margin. The pre-apical crossband is partially or fully detached from vein C, with a hyaline region distally in cell *R*_2+3_. Vein *R*_2+3_ is straight, terminating in a distinct, anteriorly inclined spur. The radio-medial crossvein intersects the discal medial cell near its midpoint. The basal medial cell is narrow and triangular, 2.5–3 times longer than wide, matching the width of cell Cup. An anal streak is absent or incomplete. Male terminalia include an elongated tergite IX lobe posteriorly, surstyli exceeding half its length, and a narrower posterior lobe in lateral view. The female ovipositor shows a typical flattened aculeus lacking serrations, accompanied by three sclerotised spermathecae [27,31,32].

### 3.2. Geographic Distribution and Spread

The Baluchistan melon fly has a broad geographic distribution across Africa, Asia, and Europe (Figure 2). It is documented in over 20 countries, including Sudan (present [33]), Egypt (present [34]), Afghanistan (present, widespread; [17,19,28,35,36]), Mainland China (intercepted only, [18,32]), India (present, [37,38,39,40,41]), Iran (present, widespread; [42,43,44,45,46,47,48]), Iraq (present, [49,50]), Israel (present, few occurrences; [51]), Jordan (present, [52]), Kazakhstan (present, [21,22]), Kyrgyzstan (present, [52]), Lebanon (present, [52]), Myanmar (present, [53]), Pakistan (present, widespread; [54,55,56]), Palestine (present, [57]), Saudi Arabia (present, [58,59]), Syria (present, [60]), Tajikistan (present, [52]), Turkmenistan (present, [61]), Uzbekistan (present, [29,62]), Armenia (present, [63]), Azerbaijan (present, [64,65]), Cyprus (present, widespread; [52,66]), Georgia (present, [52]), Russia (present, [67]), Turkey (present, widespread; [27,68,69,70,71,72,73]), Ukraine (present, [74]). The species is classified as a quarantine pest by the EU, Egypt, Mainland China, the United States, Kazakhstan, Switzerland, the United Kingdom, Ecuador, Indonesia, Japan, Peru, Thailand, and New Zealand [32,52,75,76,77]. As a result, stringent import controls are enforced in non-endemic regions to prevent the *M*. *pardalina* introduction.

A key factor contributing to *M*. *pardalina*’s invasive potential is its ability to overwinter as pupae in sub-zero, snow-prone environments [19,22,69]. This trait poses significant risks to temperate cucurbit-growing zones such as North America and Southern Europe. Dispersal occurs primarily through the movement of infested fruits harbouring larvae or pupae [69,78]. Although currently confined to Central Asia and parts of Eastern Europe, MaxEnt models predict its potential establishment globally under both current and future climatic conditions [23]. Europe and China are particularly vulnerable due to extensive host availability [23]. The pest’s accelerating range expansion highlights the urgent need for enhanced phytosanitary protocols and cross-border cooperation to safeguard non-infested regions.

### 3.3. Host Range

The Baluchistan melon fly is an oligophagous specialising in cucurbitaceous plants, infesting both cultivated and wild species. Primary cultivated hosts include melon (*C*. *melo*), with significant infestations also reported in watermelon (*C*. *lanatus*), cucumber (*C*. *sativus*), snake melon (*C*. *melo var*. *flexuosus*), and giant pumpkin (*C*. *maxima*) [29,32,79,80]. Wild hosts include *C*. *trigonus* and *Ecballium elaterium* [23,78], underscoring the pest’s adaptability to diverse ecological niches.

### 3.4. Life Cycle

The life cycle of *M*. *pardalina* encompasses four stages: Egg, larva, pupa, and adult, with the pupal stage acting as the overwintering phase [44,69]. Pupae typically reside in the soil at depths of 1–2 cm to 15–16 cm, surviving under snow cover and temperatures just below freezing [19,65,69]. Adults emerge synchronously with the melon flowering season, typically from mid-May to early June in the eastern Mediterranean [19,65,78]. Both sexes are polygamous, mating repeatedly post-emergence [43]. Females oviposit at least 100 eggs beneath the epidermis of developing fruits; upon hatching, larvae immediately tunnel into the pulp to feed [65,69]. After completing development, mature larvae exit the fruit and pupate in the soil, where they overwinter. The overwintering pupal stage presents a critical target for control measures, as interventions during this phase can significantly suppress populations in subsequent generations [32,62].

The developmental duration of *M*. *pardalina* varies with environmental conditions. Under summer field conditions, *M*. *pardalina* completes its life cycle in approximately 30 days, with eggs hatching in 2–3 days, larvae maturing in 8–18 days, and pupae developing in 13–20 days [65,81,82]. Laboratory studies under controlled conditions (25 ± 1 °C, 65% ± 5 relative humidity, 16:8 light-dark cycles) report that eggs hatch in 1.5–3.7 days, larval and pupal stages span 5–13 and 12–19.16 days, respectively [51,71]. Adults survive 10–20 days, with preoviposition and oviposition periods ranging from 2 to 6 and 11–18 days under field conditions [69,81]. These rapid developmental rates enable 2–3 overlapping generations annually in most regions [65,69,83], though up to four generations have been documented in Iran [44] and Israel [51].

### 3.5. Damage

Infestation by *M*. *pardalina* larvae causes severe damage to cucurbit fruits (Figure 3). Internal larval feeding triggers rapid tissue decay, leading to fruit rot, foul odours, and premature decomposition [68,69,78,84] (Figure 3a,b). Mature larvae exit fruits through visible holes to pupate in soil, further compromising structural integrity and rendering produce unmarketable and inedible [80] (Figure 3c).

Economic losses vary regionally but consistently threaten agricultural stability. In Israel, melon losses reach 85–90%, while watermelons suffering 60% damage [85]. Turkmenistan reports 56.7% losses in melons, alongside declines of 2.8% in watermelon, 1.1% in pumpkin, and 0.1% in cucumber; the pest’s spread has reduced melon yields by 80–90% [19]. Armenia records melon losses of 6.7–34.5% [86]. Afghanistan faces 30–40% losses in unprotected melons, with less than 5% in cucumber and watermelon [19]. In Kazakhstan’s Kyzylorda region, infestations affect 50% of farms, with losses ranging from 10 to 25%, escalating to total crop failure in severe cases [21].

During severe outbreaks, females lay eggs in unopened flowers. This enables larvae to tunnel into stems and leaf stalks before fruits form [19,69,84]. Such internal feeding makes surface-applied contact insecticides ineffective. Systemic pesticides or strategies targeting vulnerable life stages—such as overwintering pupae or emerging adults—are thus required. These measures are essential to disrupt the pest’s lifecycle and mitigate its economic and agricultural impacts.

## 4. Management

### 4.1. Monitoring and Quarantine

Effective monitoring and quarantine measures are vital for controlling *M*. *pardalina* spread. Advanced molecular detection techniques have been developed to improve detection accuracy. For example, Jiang et al. (2024) [87] introduced a visualised Loop-mediated Isothermal Amplification (LAMP) method. This identifies *M*. *pardalina* through a simple colour-change reaction, making it viable even in resource-limited settings. Complementing this, Rao et al. (2024) [88] developed a Recombinase Polymerase Amplification (RPA)-CRISPR/Cas12a detection kit. It enables rapid, on-site identification with high specificity and sensitivity at constant temperatures (37–42 °C), without requiring complex equipment. Additionally, the 2024 sequencing of the *M*. *pardalina* mitochondrial genome [18] provides insights into species diagnosis and evolutionary biology. This genomic data supports early detection and innovative controls, such as targeting genes linked to insecticide resistance or pheromone reception.

Quarantine measures include promising physical controls. Ionising radiation, such as a 65 Gy sterilisation dose, balances male survival and female sterility [22]. Temperature manipulation also shows potential. However, further research into the pest’s thermal biology is needed to refine these protocols [89].

### 4.2. Cultural Practices

Cultural and physical methods reduce *M*. *pardalina* infestations by altering environmental conditions. Adults prefer shaded areas like foliage or plant bases during peak heat [17]. Maintaining weed-free fields with ample sunlight and airflow deters pest activity. In dense plantings, repositioning fruits to sunlit areas and trimming excess foliage during early fruiting enhances canopy light penetration [19]. Proper disposal of infested fruits breaks population cycles. Traditionally, burial at 1 m depth with lime is used, but depths exceeding 50 cm may be necessary for adult mortality [78]. Additional strategies include the following: Crop rotation and early planting to disrupt pest life cycles; bagging young fruits to block oviposition—for instance, in Pakistan, this increased melon production from 2500 to 40,000 units [85]; post-harvest management, such as removing plant residues, pruning vines, and thinning fruits to an optimal density [19,20,36,42,65]. While eco-friendly, these methods face limitations under high pest pressure or in shallow-ploughed soils and may demand significant labour.

### 4.3. Chemical Control

Chemical insecticides are a key tool for managing *M*. *pardalina*. Early Soviet studies showed 0.25% Zitan-85 with Trichlorphon or Carbaryl was effective, while repellents like Phosalone, Dimethoate, and Endosulfan were less so [64]. Field trials found that straight-spray or bait-spray applications of Tamaron and E.P.N. (600 and 810 g a.i./ha every 10 days) provided superior control against larvae, followed by Padan, Sumicidin, and Dimilan. In granular applications, Thimet and Fundal (1800 and 900 g a.i./ha) outperformed Miral, Marshal, and Dacamax against larvae [54]. Laboratory experiments identified Apholate (0.1%) and Thiotepa (0.05%) as effective options [63]. In Pakistan, localised Endosulfan (3 mL/L water) or bait sprays (protein hydrolysate + Diptrex™ 80SP) applied to the 5 m periphery of melon fields reduced infestations and increased yields [55]. Similarly, in Iran, Phosalone at 35%, Trichlorfon at 80%, and Fenvalerate at 20% reduced *M*. *pardalina* populations effectively [44]. Afghanistan used Deltamethrin and carbaryl dust at infested melon removal sites to target emerging adults [19]. A 2014 Badghis study combined Diazinon, Monitor, Danadium, Laser, and Confidor with pupae removal and fruit bagging, reducing damage [28]. In Kazakhstan, sequential applications of Thiamethoxam/Cyhalothrin followed by Chlorpyrifos/Cypermethrin sustained population control and improved fruit quality for 14 days [21]. Further research in the region demonstrated that a specific treatment regimen—applying Enjio 247 SC (Thiamethoxam, 141 g/L + Lambda-Cyhalothrin, 106 g/L) at 0.25 L/ha at the end of melon flowering, followed by Nurelle D, C.E. (Chlorpyrifos, 500 g/L + Cypermethrin, 50 g/L) at 0.7 L/ha during fruit formation, and a second application of Enjio 247 SC at 0.25 L/ha during the mass emergence of the second generation of melon flies—reduced melon fruit infection by 89.0–91.3% [22]. Additionally, applying Nurelle D, C.E. at 0.7 L/ha during the same period decreased damage by 83.9–90.4%, and the yield of healthy fruits increased by 80.2–77.5 c/ha [22]. In Khorezm, Belmak 5% em.k., Detsis 2.5% em.k., and Tsipi 25% em.k. achieved over 80% efficacy, with Belmak performing best [90]. Despite efficacy, rising resistance and environmental concerns drive demand for sustainable alternatives.

### 4.4. Biological Control

Biological control may provide a sustainable approach to managing *M*. *pardalina*, with several promising methods under investigation. The sterile insect technique (SIT) involves releasing sterile males to suppress fertile offspring. While this method has been tested, its efficacy is limited by the fly’s multiple mating behaviour. However, supplementary use of sex attractants may be promising in overcoming this limitation [43]. Traditional fruit fly lures, such as Cue-lure, Methyl Eugenol, have proven ineffective in regions like Turkey and Afghanistan [19,28]. In contrast, monitoring techniques in Kazakhstan have demonstrated greater potential for surveillance. These include pheromone traps (unspecified chemical composition), yellow sticky traps, and feeding traps baited with melon juice syrup and sugar [21]. In Herat, Afghanistan, a bait comprising boiled beef, cucumber extract, and urea achieved effectiveness [35]. Tests of various baits such as melon fruit, sugars, and proteins revealed that only melon fruit consistently attracts *M*. *pardalina* [19]. Recent research has identified species-specific attractants: 4-(4-methoxyphenyl)-2-butanone and 1,4-benzyl dicarboxylate were isolated for male trapping in Uzbekistan [62], while synthesised bis(2-ethylhexyl) ester of 1,4-benzene dicarboxylic acid has enhanced monitoring efforts [29]. Additionally, laboratory trials with the entomopathogenic nematode *Heterorhabditis bacteriophora* have demonstrated efficacy against *M*. *pardalina* pupae [73], suggesting potential for future field applications. These advancements highlight the potential of biological control, yet further research is needed to refine and integrate these methods into comprehensive pest management strategies for *M*. *pardalina*.

### 4.5. Host Resistance

Host plant resistance is an important component in integrated pest management programmes. Research shows that cucurbit susceptibility to *M*. *pardalina* often correlates with physical traits, particularly thinner skins. These are more readily penetrated by the fly’s ovipositor, resulting in greater damage [78]. Genomic insights enable breeders to target specific traits such as skin thickness or biochemical resistance mechanisms. This approach facilitates the development of melon cultivars with reduced vulnerability to *M*. *pardalina* and concurrent pathogens [71]. In Kazakhstan, for example, ongoing breeding programmes are focused on developing resistant melon varieties [21]. In Iran’s Sistan region, the Sefidak and Firoozi99 melon cultivars exhibited the lowest pest damage (18–20%). Extended fruiting periods in these cultivars correlated with reduced infestation, though skin thickness exhibited no significant protective effect. Both cultivars are now prioritised in pest-resistant planting strategies [91]. Similarly, in Şükurlu, Turkey, four melon varieties—Balhan, Balözü, VT21B, and the local ‘Winter melon’ genotype ‘VN2136’—were assessed for damage by *M*. *pardalina*. All displayed damage rates below 10%, with no notable differences among them, indicating potential inherent resistance [72]. Widespread adoption of such genetically resistant varieties could deliver sustainable, long-term protection against *M*. *pardalina*. This would reduce dependence on recurrent pest control interventions while mitigating economic losses [19,86].

## 5. Discussion

Our review indicates that the Baluchistan melon fly poses a growing threat to cucurbit production across Central Asia, the Middle East, and beyond, with profound implications for agricultural economies and global food security. Capable of devastating up to 90% of yields during outbreaks, *M*. *pardalina* imposes substantial economic burdens in regions where cucurbits serve as critical cash crops. Its resilience—demonstrated through sub-zero overwintering capacity and oligophagous host specificity—combines dangerously with climate-driven range expansion and global trade networks. Without intervention, *M*. *pardalina* risks invading major cucurbit-growing zones in North America and Southern Europe, threatening livelihoods and international supply chains. Containing this threat necessitates an integrated strategy combining pest biology research, innovative control technologies, and transnational policy coordination.

Approximately two-thirds of the synthesised evidence in our review derives from the grey literature, including governmental and regional institutional reports. This reflects *M*. *pardalina*’s current concentration in Central Asia and the Middle East, where local agricultural agencies recognise it as an emerging threat. While the grey literature offers critical insights into regional priorities and practical challenges [26], its dominance highlights a stark disparity: The pest remains underrepresented in high-impact, peer-reviewed journals. This likely stems from its limited establishment in economies with strong research infrastructures, reducing incentives for global scientific engagement. Furthermore, institutional and financial constraints in affected regions may hinder researchers’ capacity to publish internationally, potentially obscuring the problem’s true scale.

The scarcity of peer-reviewed studies on *M*. *pardalina* in the global literature signals broader neglect of pests endemic to developing agricultural systems, despite their potential for cross-border proliferation. Given the pest’s capacity for severe economic losses and climate-modelled expansion potential [23], this research gap demands urgent attention. Strengthening collaborations between affected regions and international agronomic institutions could bridge knowledge divides while equitably allocating resources. Prioritising *M*. *pardalina* in global surveillance frameworks and funding initiatives would both mitigate regional vulnerabilities and pre-empt future crises as trade and climate patterns evolve.

Future management efforts should adopt a holistic integrated pest management (IPM) framework for *M*. *pardalina*, combining cultural, chemical, biological, and genetic strategies. Cultural tactics, such as optimising planting schedules and field layouts to disrupt the pest’s life cycle through enhanced sunlight exposure and airflow, require systematic evaluation. Transition from broad-spectrum insecticides [89,92] to precision technologies, including drone-targeted applications [93], bioinformatics-driven compound discovery [94], and AI-enabled monitoring systems [95]. While parasitoids, entomopathogens, and nematodes are widely used against other fruit flies [96], *M*. *pardalina*’s known natural enemies remain limited to three ant species (*Cataglyphis bicolor*, *C*. *megalocola*, and *Pheidole pallidula*) that prey on larvae [83]; their field efficacy, however, remains unquantified. Although *H*. *bacteriophora* achieved 80% pupal mortality in laboratory trials [73], its field applicability demands validation. Expanding biocontrol exploration to include parasitoid wasps and fungi is critical. Concurrently, refining attractants—such as pheromonal and food attractants [21,29,62]—requires deeper insights into the pest’s chemical ecology to improve scalability [97,98,99,100]. Genomic advances, including CRISPR-Cas9 gene editing [101] and RNA interference [102] informed by mitochondrial sequencing [18], could disrupt pest reproduction or accelerate resistant crop development. Climate modelling to predict range expansion under warming scenarios should inform pre-emptive containment strategies.

Policymakers should act decisively to translate research into actionable measures, preventing *M*. *pardalina* from becoming a global crisis. National governments should fund interdisciplinary projects bridging laboratory innovations with farm-level solutions, potentially through subsidies for resistant cultivars and IPM certification schemes. International bodies, including the European and Mediterranean Plant Protection Organisation (EPPO) and the Food and Agriculture Organisation (FAO), should standardise quarantine protocols, facilitate germplasm exchange, and coordinate transnational monitoring efforts. Equally critical is enhancing agricultural extension services to train smallholders in IPM techniques, ensuring access to pheromone traps and climate-resilient seeds. Structured collaboration between researchers, policymakers, and farming communities offers the most viable path to mitigate economic losses, safeguard food security, and build long-term resilience against this escalating threat.

## Figures and Tables

**Figure 1 insects-16-00514-f001:**
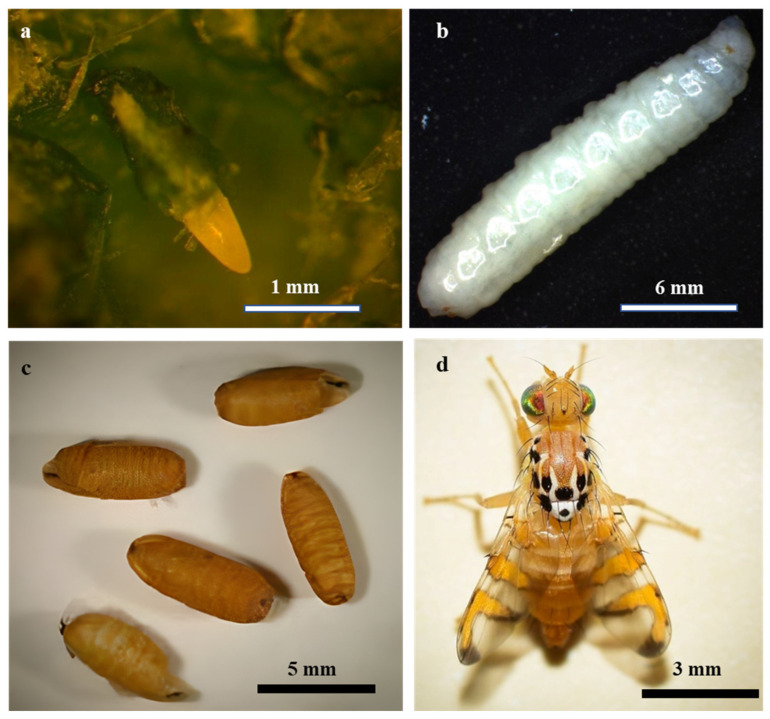
Life stages of *M*. *pardalina*: (**a**) egg (adapted from Baris and Cobanoglu [27]), (**b**) larva (adapted from Baris and Cobanoglu [27]), (**c**) pupa (adapted from Kholbekov et al. [29]), (**d**) adult (adapted from Ruslan Mishustin).

**Figure 2 insects-16-00514-f002:**
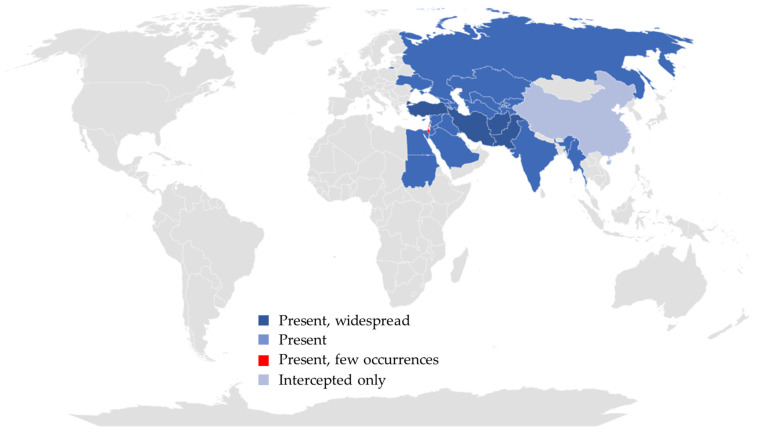
Geographic distribution of *M*. *pardalina*.

**Figure 3 insects-16-00514-f003:**
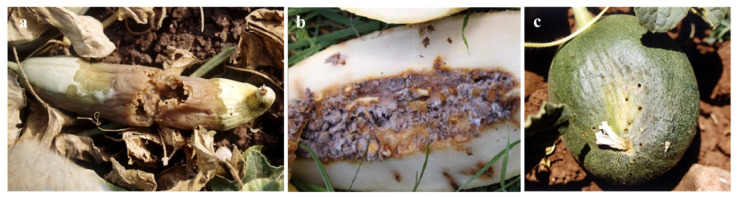
Fruit damage by *M*. *pardalina* larvae (adapted from [78] and Biochemtech, https://biochemtech.eu/products/melon-fly-myiopardalis-pardalina (accessed on 18 March 2025)): (**a**) decay of fruit tissue with visible rot symptoms, (**b**) internal tissue maceration and decomposition, (**c**) larval damage showing exit holes.

## Data Availability

No new data were created or analysed in this study.

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
