# Peer review of "The Baluchistan Melon Fly, Myiopardalis pardalina Bigot: Biology, Ecology, and Management Strategies"

_insects, 2025, doi:10.3390/insects16050514_

Round 1

Reviewer 1 Report

Comments and Suggestions for Authors

The authors have reviewed information about a pest that causes significant damage to a major group of crops. 

-The parentheses in the scientific name in the title should be removed. Please add Author name of the fly. 

-Line 28-30 The sentence needs to be broken down. While it refers to the life cycle and morphology, it also discusses the spread, which makes it hard to follow.

-In line 35, it is better to use 'physical control' instead of 'physical interventions' and 'chemical management' instead of 'chemical insecticides.

-line 54- please use synonym of "exemplifies" and "jeopardises" in the sentence.

-It would be more appropriate for the second paragraph to start with line 56 ('melon...') and for the sentence beginning with 'Baluchistan fly' (line 54) to appear at the end of the paragraph.

-Line 162 Life cycle should be reviewed under a separate heading

-line 187 please rewrite the headin

-line 212 please use "Management" instead of "control".

-Line 231: Please use 'Cultural practices' or 'Cultural measures' instead of 'cultural control.

Comments on the Quality of English Language

The English in the text needs to be reviewed. The sentences are long, and different subjects are introduced while one is being discussed. The sentences should be divided and shortened. More precise, field-specific, and commonly used scientific terminology should be chosen

Reviewer 2 Report

Comments and Suggestions for Authors

Dear Authors,
I have carefully read your manuscript regarding the review of the fruit fly Myiopardalis pardalina. The review is accurate and I have only a few corrections that I write below; only one suggestion that I consider important: it is to transform the table relating to the distribution of the diptera into a detailed map.

Table 1 : in my opinion it could be better to replace this table with a map

Line 188: replace catastrophic with severe damage

Lines 126-127: please use correct term for the ovipositor such as aculeus; e.i. “the female shows a typical flattened aculeus etc”

Lines 252-253: not clear sentence, please rewrite since is not so clear that granular applications are against what kind of preimaginal stage: perhaps pupae in the soil?

Round 2

Reviewer 1 Report

Comments and Suggestions for Authors

Accept in present form